# Clinical Outcomes Following Re-Operations for Intracranial Meningioma

**DOI:** 10.3390/cancers13194792

**Published:** 2021-09-24

**Authors:** George E. Richardson, Conor S. Gillespie, Mohammad A. Mustafa, Basel A. Taweel, Ali Bakhsh, Siddhant Kumar, Sumirat M. Keshwara, Tamara Ali, Bethan John, Andrew R. Brodbelt, Emmanuel Chavredakis, Samantha J. Mills, Chloë May, Christopher P. Millward, Abdurrahman I. Islim, Michael D. Jenkinson

**Affiliations:** 1Institute of Systems, Molecular & Integrative Biology, University of Liverpool, Liverpool L69 7BE, UK; c.s.n.gillespie@liverpool.ac.uk (C.S.G.); m.a.mustafa@liverpool.ac.uk (M.A.M.); b.taweel@liverpool.ac.uk (B.A.T.); ali.bakhsh@doctors.org.uk (A.B.); siddhant.kumar@thewaltoncentre.nhs.uk (S.K.); hlskeshw@liverpool.ac.uk (S.M.K.); christopher.millward@thewaltoncentre.nhs.uk (C.P.M.); abdurrahman.islim@thewaltoncentre.nhs.uk (A.I.I.); michael.jenkinson@liverpool.ac.uk (M.D.J.); 2Department of Neurosurgery, The Walton Centre NHS Foundation Trust, Liverpool L9 7LJ, UK; tamara.ali@thewaltoncentre.nhs.uk (T.A.); bethan.john@nhs.net (B.J.); abrodbelt@doctors.org.uk (A.R.B.); emmanuel.chavredakis@thewaltoncentre.nhs.uk (E.C.); 3Department of Neuroradiology, The Walton Centre NHS Foundation Trust, Liverpool L9 7LJ, UK; Samantha.Mills@thewaltoncentre.nhs.uk; 4Department of Clinical Oncology, Clatterbridge Cancer Trust, Liverpool CH63 4JY, UK; chloemay2@nhs.net

**Keywords:** meningioma, re-operations, outcomes, recurrence, complications, surgery

## Abstract

**Simple Summary:**

This study investigated patients who underwent more than one operation for a meningioma, a type of brain tumor. Currently, there is little evidence available for this specific patient group. The purpose of this study was to determine if patients had an improvement or deterioration following a second operation for a recurrent meningioma, and to identify any factors that may influence this change. The results demonstrated that following a second operation for meningioma, patients have poorer outcomes. The findings of this study provide supporting information for surgeons and patients, thereby informing decisions related to patient care and re-operation.

**Abstract:**

The outcomes following re-operation for meningioma are poorly described. The aim of this study was to identify risk factors for a performance status outcome following a second operation for a recurrent meningioma. A retrospective, comparative cohort study was conducted. The primary outcome measure was World Health Organization performance. Secondary outcomes were complications, and overall and progression free survival (OS and PFS respectively). Baseline clinical characteristics, tumor details, and operation details were collected. Multivariable binary logistic regression was used to identify risk factors for performance status outcome following a second operation. Between 1988 and 2018, 712 patients had surgery for intracranial meningiomas, 56 (7.9%) of which underwent a second operation for recurrence. Fifteen patients (26.8%) had worsened performance status after the second operation compared to three (5.4%) after the primary procedure (*p* = 0.002). An increased number of post-operative complications following the second operation was associated with a poorer performance status following that procedure (odds ratio 2.2 [95% CI 1.1–4.6]). The second operation complication rates were higher than after the first surgery (46.4%, *n* = 26 versus 32.1%, *n* = 18, *p* = 0.069). The median OS was 312.0 months (95% CI 257.8–366.2). The median PFS following the first operation was 35.0 months (95% CI 28.9–41.1). Following the second operation, the median PFS was 68.0 months (95% CI 49.1–86.9). The patients undergoing a second operation for meningioma had higher rates of post-operative complications, which is associated with poorer clinical outcomes. The decisions surrounding second operations must be balanced against the surgical risks and should take patient goals into consideration.

## 1. Introduction

Meningiomas account for 38.3% of primary brain tumors [1,2]. Their presentation is variable and 30% of tumors are discovered incidentally [3,4,5]. Meningiomas are graded using the World Health Organization (WHO) classification [4]; Grade 1 represents 81% of meningiomas and are the most indolent, with grades 2 and 3 being incrementally more aggressive and uncommon [1,3]. The mainstay in treatment for a symptomatic meningioma is surgery [4,6]. The Simpson Grading System is used to classify the extent of resection [7] and is a key prognostic factor for recurrence [8]. Other prognostic factors for recurrence include the WHO tumor grade and use of adjuvant radiotherapy [4,9]. 

There are multiple treatment strategies for patients who suffer recurrence following an initial resection. These include active monitoring, further surgery, fractionated radiotherapy (*f*RT), or stereotactic radiosurgery (SRS) [4,10]. The available literature for outcomes following re-operation is sparse [11,12,13]. The re-operation for recurrence is associated with high rates of post-operative complications, for both skull base and non-skull base tumors [11,12]. The risk factors include posterior fossa, middle third parasagittal location, and pre-reoperation cognitive changes [11,12]. Additionally, with each re-operation, the time between surgeries decreases [13]. With increased morbidity and reduced “time bought” from each successive surgery, it is important to balance the risks versus benefits. There are no studies assessing performance status outcomes for patients following re-operation for a meningioma.

The primary aim of this study was to investigate factors associated with changes in performance status following a second operation for a recurrent meningioma. The secondary aims were to determine complication rates and overall progression free survival.

## 2. Materials and Methods

### 2.1. Study Design and Population

This was a retrospective cohort study of patients that underwent a second operation for the recurrence of an intracranial meningioma between 1 January 1988 and 31 December 2018. Patients treated with reoperation for recurrence of the same intracranial meningioma at a tertiary neuroscience center were included. Patients with extracranial meningioma, staged resections, other treatment prior to initial surgical resection, or repeat operation for separate meningiomas were excluded [14]. A follow up was included up to 31 December 2020. A comparative analysis of patients undergoing only one operation during the same time period was conducted. The study was approved by hospital audit departments. This manuscript was produced according to the Strengthening the Reporting of Observational Studies in Epidemiology statement [15]. 

### 2.2. Study Outcomes

#### 2.2.1. Primary Outcome

The primary outcome measure was the WHO performance status following a second operation [16]. The performance status was recorded at routine follow-up visits, and for the purpose of this study, was recorded six months after the second operation.

#### 2.2.2. Secondary Outcomes

Post-operative complications within 30 days were recorded and classified according to the Landriel-Ibañez and Clavien-Dindo classifications (Appendix A) [17,18]. The persistent complications after the first operation that were present after the second operation were not re-recorded unless there was a worsening of the condition. Additionally, 30-day post-operative mortality data was collected. Tumor recurrence was recorded and defined using the Response Assessment in Neuro-Oncology (RANO) criteria [19]. These criteria include patient clinical status and tumor volume throughout follow-up. A comparison between re-operated and non-re-operated patients was carried out for the extent of tumor resection (according to Simpson grade), the WHO tumor grade, adjuvant treatment, and progression (according to RANO criteria) [7,19]. 

### 2.3. Clinical and Radiological Variables at Primary and Repeat Surgery

Clinical characteristics were collected and recorded at the time of the primary surgery. Patient sex, age, presenting clinical features, the WHO performance status, and Age-Adjusted Charlson Comorbidity Index (ACCI) were included [16,20]. The risk factors associated with meningioma development, including a history of cranio-spinal radiation, hormone replacement therapy, and genetic syndromes were recorded [3,21,22]. The radiological features of the meningioma before the first operation were recorded. The tumor location was defined according to the International Consortium on Meningiomas (ICOM) classification [23]. The meningioma volume was calculated using the simplified ellipsoid formula (ABC/2) and reported in centimeters cubed (cm^3^) [24,25]. The inter- and intra-rater reliability of volume measurement was assessed by calculating the intraclass correlation coefficient (ICC). This was performed on repeated measurements by the primary researcher (intra-rater) and by two blinded researchers (inter-rater). The meningioma signal intensity was categorized into hyperintense and iso/hypointense. Hyperintensity was defined as increased signal-change in the meningioma compared to the contralateral gray matter, on T2 weighted or fluid-attenuated inversion recovery (FLAIR) magnetic resonance imaging (MRI) [5]. The presence of peri-tumoral signal change (indicative of vasogenic oedema) was collected using T2 weighted or FLAIR MRI [5]. The meningioma multiplicity was recorded, and when present, only the details of the tumor subject to the second operation were collected. The clinical and imaging details related to each surgery were collected. These included the extent of resection as per the Simpson Grading Scale, presence and volume of the residual tumor identified on a post-operative MRI, management strategy for a residual tumor, tumor grade, and the WHO performance status pre-surgery and six months post-operatively [4,7,16]. Adjuvant treatment was defined as additional therapy given within six months of an operation, and data on radiotherapy fractions and doses were recorded. 

### 2.4. Comparative Cohort

A comparative cohort of meningioma patients treated with a single operation was generated from the overall database. For the patients that underwent a single operation, the following variables were collected: the Simpson grade of resection, the WHO grade [4,7], recurrence, and use of adjuvant radiotherapy.

### 2.5. Statistical Analysis

A statistical analysis was conducted using SPSS version 26 (IBM Corp.). The R version 4.1.0 was used for graphical analysis. The continuous variables were subject to a Kolmogorov-Smirnov test of normality. The normally distributed variables are presented using mean and standard deviation (SD); the skewed variables are presented using median and interquartile range (IQR). Descriptive statistics were used to compare variables between the re-operated and non-re-operated cohorts. A univariate analysis was conducted using binary logistical regression to identify factors associated with worsened performance status following a second operation. The variables with a *p*-value ≤ 0.2 were included in the multivariable model. A multivariable binary logistic regression model was used to identify risk factors, and a *p*-value ≤ 0.05 was considered statistically significant. The Kaplan Meier analysis was used to assess overall and progression free survival, along with log rank tests where applicable. Chi-Squared tests were used to assess the differences in categorical variables. 

## 3. Results

### 3.1. Re-Operated Study Population and Baseline Clinical and Imaging Features

Between 1988 and 2018, 712 meningioma patients were treated with surgery, 56 (7.8%) of which had a second operation for recurrent meningioma. The mean age at initial diagnosis was 50.1 years (SD = 10.7, range 30–73), and the mean age at the second operation was 56.1 years (SD = 11.6, range 30–77). Two patients (3.6%) had a history of previous cranio-spinal radiation and no other risk factors were identified. The most common presenting symptom was a headache (50%, *n* = 28). The most common tumor location at diagnosis was convexity (33.9%, *n* = 19), and peri-tumoral signal change was identified in 19 patients (33.9%). Tumor hyperintensity was identified in 20 (35.7%) patients. The mean tumor volume at diagnosis was 38.2 cm^3^ (SD = 32.6). Multiple meningiomas were identified in eight (14.3%) patients. Table 1 shows the baseline patient demographics, and Figure 1 details the treatments that each patient received. The inter- and intra-rater reliability of tumor volume measurement was adequate (ICC 0.93 and 0.97, respectively). Appendix A contains full ICC results. 

### 3.2. Cohort Comparison: Re-Operated versus Non-Re-Operated

The proportion of Simpson grade 4 resections following a first operation was lower in the 656 non-re-operated patients compared to the re-operated cohort (*n* = 124, 18.9% versus *n* = 31, 55.4%). There were 200 (30.5%) Simpson grade 1, 248 (37.8%) Simpson grade 2, and 30 (4.6%) Simpson grade 3 resections in the non-re-operated group. Simpson grades following a first operation for the re-operated cohort are presented in Table 2. There were 492 (75.0%) WHO grade 1 and 199 (30.3%) WHO grade 2 tumors. There were no WHO grade 3 tumors in the non-re-operated cohort. From this group, a total of 32 patients underwent adjuvant radiotherapy (4.9%), three (0.5%) had SRS, and 29 (4.4%) had fRT. Following the first operation in the re-operated cohort, there were 29 (51.8%) grade 1, 25 (44.6%) grade 2, and one (1.8%) grade 3 meningiomas. In the non-re-operated cohort, 98 (14.9%) patients demonstrated tumor progression according to the RANO criteria. A further 15 (2.3%) patients demonstrated clinician reported progression, which did not satisfy the RANO criteria.

### 3.3. Primary Resection and Adjuvant Treatment in Re-Operated Patients

On a post-operative MRI, residual tumors were present in 55.4% of cases (*n* = 31). The median residual tumor volume was 1.9 cm^3^ (IQR = 0.5–4.6). Adjuvant radiation was administered to seven (12.5%) patients (WHO grade 2, *n* = 6, WHO grade 3, *n* = 1). Three patients received 54Gy in 30 fractions, and four received 60Gy in 30 fractions. The surgical details for the first operation in the re-operated cohort are included in Table 2. 

### 3.4. Recurrence and Indications for Second Operation

The median time to the first recurrence was 35.5 months (IQR = 18.5–71.5), with 71.4% recurring radiologically (*n* = 40) and 21.4% recurring radiologically and clinically (*n* = 12). One patient recurred only clinically, having developed worsening symptoms with no changes to the residual tumor on an MRI. For three patients, there was incomplete data due to the historic nature of the cases. Of the 40 patients with tumor recurrence on MRI, 62.5% (*n* = 25) fulfilled RANO criteria for radiological progression. All patients with symptomatic progression fulfilled the RANO criteria. The more common symptoms at recurrence were optic neuropathy (66.6%, *n* = 8) and headaches (25%, *n* = 3). The management of meningioma recurrence after a primary operation included surgery (*n* = 33, 58.9%), surgery and *f*RT (*n* = 14, 25%), SRS (*n* = 4, 7.1%), surgery and SRS (*n* = 3, 5.4%), and *f*RT (*n* = 2, 3.6%).

### 3.5. Second Operation Clinical Details

At the second operation, 32.1% (*n* = 18) achieved a Simpson grade 1 or 2 resection. The majority of tumors were grade 1 (48.2%, *n* = 27), while grade 2 and 3 tumors accounted for 44.6% (*n* = 25) and 5.3% (*n* = 3), respectively. Following a second operation, 10 patients (17.9%) had a change in the WHO tumor grade; three decreased to a lower WHO grade, while seven increased to a higher grade. The residual tumors were present in 34 cases (60.7%). The median residual volume after a second operation was 0.9 cm^3^ (IQR = 0.4–4.0). Adjuvant radiation was administered to 23 patients (WHO grade 1, *n* = 8, WHO grade 2, *n* = 12, WHO grade 3, *n* = 3). Two patients had SRS (12.5Gy to the 80% isodose). The remaining 21 patients received adjuvant fRT over 30 fractions; 11 patients were given 54Gy and 10 were given 60Gy. Further details are included in Table 2. 

### 3.6. Comparison of Outcomes after First and Second Surgery in the Re-Operated Cohort

#### 3.6.1. Performance Status and Predictors of Outcome

The performance status decreased in more patients following the second operation when compared to the first (26.8%, *n* = 15 versus 5.3%, *n* = 3, *p* = 0.002). Figure 2 demonstrates the changes in performance status at various time points during surgical treatment.

Following the univariate analysis for factors associated with poorer performance status after a second operation, a headache at recurrence (*p* = 0.20), time to first recurrence (*p* = 0.02), and number of post-operative complications (*p* = 0.01) were included in the multivariable model. The multivariable analysis demonstrated that an increased number of post-operative complications (OR 2.2 [95% CI 1.1–4.6], *p* = 0.03) and increased time to recurrence (OR 1.01, [95% CI 1.0–1.03], *p* = 0.04) were associated with a deterioration in performance status after a second operation (Appendix A). 

#### 3.6.2. Medical and Surgical Complications

Following the first operation, 32.1% of patients (*n* = 18) in the re-operated cohort had a complication. A CSF leak was seen in five cases (8.9%) and two patients suffered a post-operative stroke (3.6%). The complication rates following a second operation were higher than those following a first surgery (48.2%, *n* = 27 versus 32.1%, *n* = 18, *p* = 0.069). After a second operation, four patients had a CSF leak (7.1%), while three patients suffered a post-operative hemorrhage (5.4%). The frequency and grade of both Landriel-Ibañez and Clavien-Dindo complications following a first and second operation are presented in Figure 3. The specific complications following a first and second surgery are listed in Table 3. 

### 3.7. Progression Free Survival and Requirement for Further Interventions

The median progression free survival following the first surgery was 35.0 months (95% CI 28.9–41.1), and 68.0 months (95% CI 49.1–86.9) following a second surgery. There was a statistically significant difference between the median PFS of patients when comparing the first and second surgery (*p* = 0.006) (Figure 4). After the second surgery, 27 patients had another recurrence at a median time of 30.0 months (IQR 21.0–57.5). Of these, 77.7% were classed as a recurrence according to the RANO criteria (*n* = 21). The median tumor volume at the recurrence was 1.5 cm^3^ (IQR 1–6.6). The most common presenting symptom at the time of the second recurrence was cranial nerve palsy (14.8%, *n* = 4); there were three cases of optic neuropathy and one case of oculomotor nerve palsy. The second recurrence meningiomas were treated with surgery (*n* = 13, 48.1%), fRT (*n* = 7, 25.9%), SRS (*n* = 3, 11.1%), and surgery with fRT (*n* = 2, 7.4%). After a second recurrence, 12 more patients had a third, four patients had a fourth, three had a fifth, and one patient had a sixth recurrence. Eleven patients had three surgeries and 3 patients had four surgeries (Figure 1).

### 3.8. Follow-Up and Overall Survival

At the time of the study’s completion, 39 patients remained under follow-up (69.6%) and 10 patients had died (17.9%). Eight of the deaths were attributed to disease progression. The majority of patients under follow-up were monitored clinically with interval MRI surveillance (76.9%, *n* = 30). The median follow up time was 128.5 months (IQR = 73–194.5). The median OS was 312 months (95 % CI 257.8–366.2) (Figure 4).

## 4. Discussion

In this study, over a period of 30 years, out of 712 patients treated surgically for intracranial meningioma, only 56 patients underwent second surgery for a recurrence. A significant worsening of performance status after a second surgery was observed compared to the first operation. The complication rates were higher following a second operation. A multivariable analysis identified increased complications and a longer time to a first recurrence as significant factors associated with a poorer performance status outcome. The median overall survival was 312 months, and although a second operation improved the control of the meningioma, 26.8% of patients had a poorer performance status following the procedure. 

### 4.1. Meningioma Recurrence

The known factors associated with meningioma recurrence have the greatest influence over the likelihood of requiring a second operation and adjuvant treatment. Meningioma recurrence following a gross total resection is uncommon, and undergoing a second operation for a recurrent tumor is equally rare [26,27]. The rates of multiple operations have been reported between 6.8% and 7.8%; these figures are concordant with the rates in this study (7.8%) [11,12,13]. The patient age, skull base location, and WHO tumor grade are factors that are associated with initial tumor recurrence [13]. Following the second operation, the skull base location and WHO grade are associated with further recurrence [13]. In this present study, there was a sizable difference in the rates of Simpson grade 4 resections and WHO grade 2 tumors between re-operated and non-re-operated patients. Counterintuitively, a greater proportion of patients were treated with adjuvant radiation after the first procedure in the re-operated cohort, which may be explained by a greater proportion of WHO grade 2 and 3 tumors. Ensuring a maximal surgical resection and the appropriate use of adjuvant treatments are the most important clinically modifiable factors that influence the likelihood of requiring repeated operations for a meningioma recurrence.

### 4.2. Complication Rates Following Second Operation

The risk profile associated with undergoing a second operation for intracranial meningioma is not comparable to the first. The complication rates following a second operation for meningioma were reported to range from 30% and 47.7%, depending on anatomical location [11,12]. Additionally, the incidence of new-onset neurological deficit is high following a second operation [13]. The meningioma location in the sagittal plane and cognitive impairment at diagnosis for non-skull base tumors and posterior fossa skull base tumors are associated with an increased risk in complications following a second operation [11,12,28]. The rates of complication following a first and second operation in this study were high (32.1% and 46.4%, respectively). The classification systems used in this study define complications as any deviation from the normal post-operative course within 30 days post-operatively [17,18]. This may account for the high complication rate following the first operation, since non-neurological complications were also recorded. Following the second operation, higher rates of neurological impairment and serious surgical complications (e.g., hydrocephalus, hemorrhage, and intracranial infection) were reported. This high rate of complications may be explained by the technical challenge of repeated surgery, characterized by post-treatment gliosis and less well-defined tissue planes [29]; this should be acknowledged when counselling patients about the risks of a second operation [30,31]. 

### 4.3. Impact on WHO Performance Status

This study is the first to investigate functional performance status outcomes following repeated operations for a recurrent intracranial meningioma. WHO performance status provides a clinical perspective on the functional capabilities of patients, with a degree of correlation to patient quality of life [32]. The patients are significantly more likely to have a deterioration in performance status after a second operation, compared to the initial procedure. Following a second operation, patients do not get the same improvement that is noted after the first procedure. The post-operative complications were significantly associated with a poorer performance status after a second operation for meningioma. The presence of a serious surgical complication, such as hydrocephalus and hemorrhage, after a second operation explains the association between complications and poorer outcomes. The increased time to recurrence was also implicated in poorer performance status following a second operation. The patients with a longer time to recurrence would be older at the time of their second surgery, and with less ‘brain resilience’ and capacity for neurological recovery; this may explain the deterioration [33]. In the context of high complication rates and subsequent poorer performance status, surgeons should carefully consider the risk to benefit ratio of a second operation for recurrence, especially in the absence of symptoms that are affecting quality of life and neurological function.

### 4.4. Benefits of a Second Operation

Despite the risks associated with a second operation, there is a quantifiable benefit to pursuing repeated resection. Previous studies have demonstrated that re-operated patients tend to have excellent overall survival rates [11,12]. The results of this current study are concordant, demonstrating a median overall survival of 26 years from diagnosis. Moreover, a significant improvement was shown between the median progression free survival following the first and second operations. This is because a proportion of patients did not have further tumor recurrence after a second operation. The improvement in tumor control may be attributable to surgeons aiming for more extensive resections when re-operating for a recurrent meningioma [34]. Although this demonstrates an apparent benefit of a second operation, the potential for a high burden of morbidity should not be ignored [35]. Moreover, the time between each subsequent operation decreases, resulting in a reduced recurrence free survival, while the risks associated with each procedure are maintained [13]. The decision-making process surrounding reoperation is multifactorial and complex. To achieve optimal management, patient choice should be heavily prioritized when making joint treatment decisions. 

### 4.5. Limitations

This study had several limitations. Firstly, undergoing a second operation for recurrent meningioma is a low frequency event; this cohort included only 56 out of 712 meningioma patients over a 30-year period. Secondly, there was a high degree of treatment strategy heterogeneity. Thirdly, WHO tumor grade was classified based on the WHO criteria in use at the time of surgery; this may result in an under-reporting of grade 2 meningiomas. Finally, Ki-67 may be an explanatory variable for changes in progression free survival [36]. However, this is not part of the current WHO classification criteria for meningioma, and there was insufficient data in our cohort for a meaningful analysis. When these factors are combined, it may limit the generalizability of the results, which may be further compounded by the retrospective nature of the study.

## 5. Conclusions

The decision to pursue a second operation for recurrent intracranial meningioma should be made in the context of patient symptoms, extent of the meningioma as shown on an MRI, WHO grade, and patient-oriented care goals. Caution is advised when immediately offering repeat surgical treatment for a patient with radiological recurrence in the absence of any change in neurological symptoms. A period of monitoring should also be considered to assess the tempo of any recurrence. The choice of treatment for recurrence should be discussed within a multi-disciplinary team with an acknowledgement of the potential increased risk in post-operative complications and worsened long term performance status. Additional large-scale, multi-center studies are needed to delineate the outcomes following repeat operation in recurrent meningiomas.

## Figures and Tables

**Figure 1 cancers-13-04792-f001:**
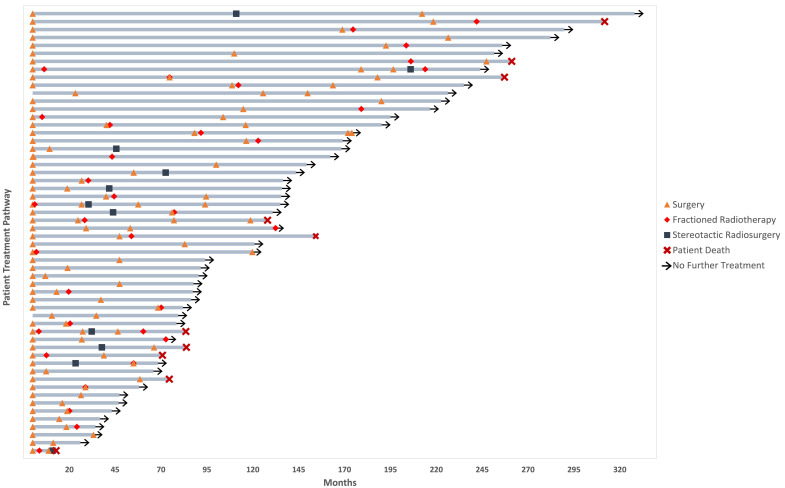
Swimmer Plot showing the treatment pathways for each patient in the cohort. The key is provided to the right. Individuals with no surgery at the beginning of treatment strategy were incidental, with primary and subsequent operations occurring at a later point during follow-up.

**Figure 2 cancers-13-04792-f002:**
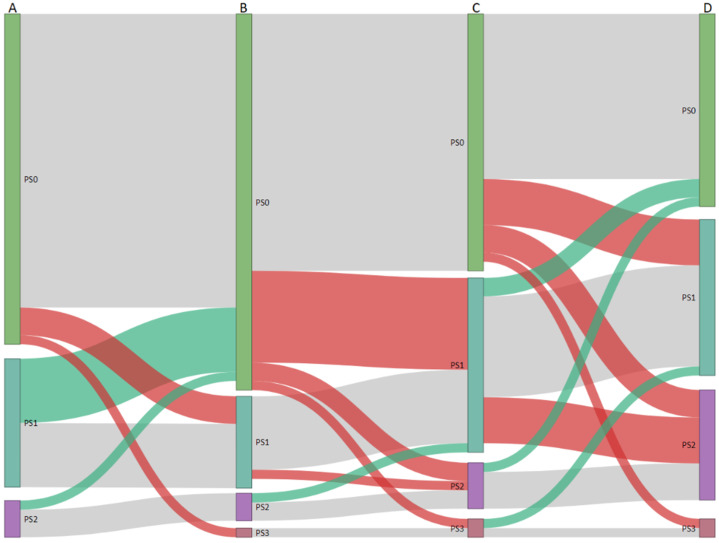
Sankey diagram demonstrating changes in performance status from initial diagnosis (A), after primary operation (B), before re-operation (C), and 6 months after re-operation (D). The length of the nodes and links between nodes are proportional to the number of patients. Red links indicate a worsening of performance status. Green links indicate improved performance status. Grey links indicate no change in performance status.

**Figure 3 cancers-13-04792-f003:**
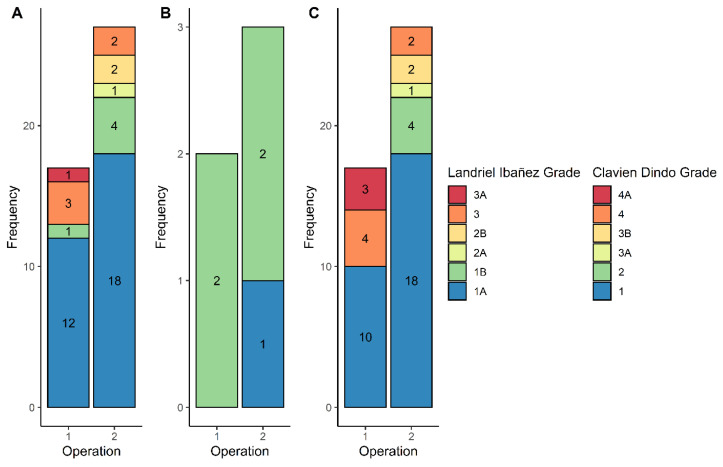
Stacked bar charts demonstrating post-operative complications graded according to Landriel-Ibañez classification (**A**,**B**) and Clavien-Dindo classification (**C**). The Landriel-Ibañez classification includes both surgical (**A**) and medical (**B**) complications. Frequency of each grade is demonstrated within the bars.

**Figure 4 cancers-13-04792-f004:**
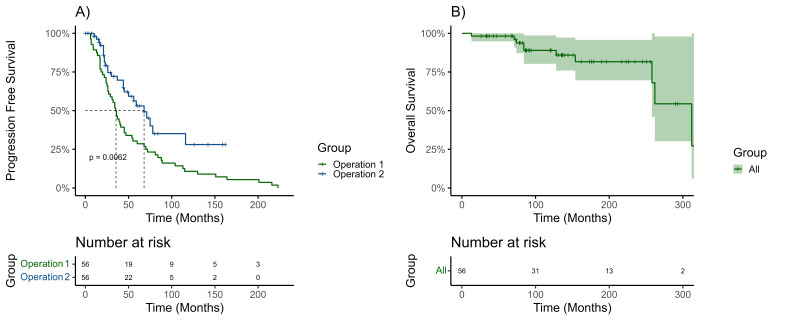
Kaplan Meier graphs demonstrating progression free survival with a log rank test between operation one and operation two (**A**); percentage at risk table is provided. Overall survival is demonstrated along with 95% confidence intervals (**B**). Censoring on both graphs is demonstrated by crosses on the curve.

**Table 1 cancers-13-04792-t001:** Contains baseline clinical and radiological details for the cohort at diagnosis.

Clinical Baseline and Imaging Characteristics	Frequency/Value
Female (%)	37 (66.0)
ACCI at Diagnosis (%)	
0	20 (35.7)
1	14 (25.0)
2	14 (25.0)
3	4 (7.1)
4	2 (3.6)
5	1 (1.8)
11	1 (1.8)
Presenting Symptoms (%)	
Incidental	3 (5.4)
Seizure	11 (19.6)
Headache	28 (50.0)
Nausea	6 (10.7)
Vomiting	4 (7.1)
Limb Sensory Changes	5 (8.9)
Limb Weakness	8 (14.3)
Cranial Nerve Deficit:	18 (32.1)
(CN I) Olfactory Nerve	4 (7.1)
(CN II) Optic Nerve	13 (23.2)
(CN III) Oculomotor Nerve	2 (3.6)
Other CNs	2 (3.6)
Expressive Dysphasia	2 (3.6)
Cognitive Deficit	15 (26.8)
Altered GCS	2 (3.6)
Other	8 (14.3)
Skull Base (%)	22 (39.3)
Sphenoid wing	8 (14.3)
Anterior Midline	10 (17.9)
Posterior Fossa–Midline	1 (1.8)
Posterior Fossa–Lateral/Posterior	3 (5.4)
Non-Skull Base (%)	34 (61)
Convexity	19 (33.9)
Parasagittal	5 (8.9)
Parafalcine	7 (12.5)
Tentorial	1 (1.8)
Intraventricular	1 (1.8)
Intraosseous	1 (1.8)

**Table 2 cancers-13-04792-t002:** Contains clinical and operative details for the first and second operations.

Clinical and Operative Variables	Operation 1 Frequency/Value	Operation 2 Frequency/Value
Pre-Operative Performance Status (%)		
0	35 (62.5)	27 (48.2)
1	14 (25.0)	18 (32.1)
2	4 (7.1)	5 (8.9)
3	0	2 (3.5)
Post-Operative Performance Status (%)		
0	39 (69.6)	21 (37.5)
1	10 (17.9)	17 (30.4)
2	3 (5.4)	12 (21.4)
3	1 (1.8)	2 (3.5)
Worsened Performance Status (%)	3 (5.4)	15 (26.8)
Simpson Grade (%)		
1	14 (25)	11 (19.6)
2	6 (10.7)	7 (12.5)
3	0	2 (3.6)
4	31 (55.4)	34 (60.7)
Management Strategy of Residual (%)		
Observation	24 (77.4)	20 (58.8)
SRS	5 (16.1)	2 (5.9)
*f*RT	2 (6.5)	12 (35.3)

**Table 3 cancers-13-04792-t003:** Post-operative complications identified after a first and second operation. Frequency of complications may be greater than the total of the category (e.g., surgical, medical, neurological complications) due to patients having had multiple complications.

Complication	Frequency Following Operation 1 (%)	Frequency Following Operation 2 (%)
Any Cause Complication	18 (32.1)	27 (46.4)
Surgical Complication	9 (16.1)	10 (17.9)
Hemorrhage	1 (1.8)	3 (5.4)
Hydrocephalus	0	3 (5.4)
SSI Incisional	2 (3.6)	3 (5.4)
SSI Intracranial	1 (1.8)	2 (3.6)
Stroke	2 (3.6)	0
CSF Leak	5 (8.9)	4 (7.1)
Pseudomeningocele	2 (3.6)	2 (3.6)
Oedema	0	1 (1.8)
Neurological Impairments	11 (19.6)	19 (33.9)
Seizures	1 (1.8)	1 (1.8)
Headache	0	1 (1.8)
Limb weakness	5 (8.9)	7 (12.5)
Limb sensory deficit	0	1 (1.8)
Cranial nerve deficit	6 (10.7)	8 (14.3)
CNI	2 (3.6)	0
CNII	2 (3.6)	4 (7.1)
CNIII	0	3 (5.4)
CNIV	0	1 (1.8)
CNV	2 (3.6)	0
CNVII	1 (1.8)	0
Language deficit	1 (1.8)	2 (3.6)
Cognitive deficit	3 (5.4)	2 (3.6)
Altered GCS	0	1 (1.8)
Personality Change	1 (1.8)	0
Ataxia	0	1 (1.8)
Medical Complication	2 (3.6)	3 (5.4)
Deep Vein Thrombosis	1 (1.8)	0
Anemia	1 (1.8)	0
Gastrointestinal Infection	0	2 (3.6)
Hyponatremia	0	1 (1.8)

## Data Availability

The data presented in this study are available on request from the corresponding author. The data are not publicly available due to containing confidential anonymized patient information.

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
