# Peer review of "Clinical Outcomes Following Re-Operations for Intracranial Meningioma"

_cancers, 2021, doi:10.3390/cancers13194792_

Round 1

Reviewer 1 Report

I read with great interest the current paper which aims to evaluate the functional performance status outcomes following repeat operation for recurrent intracranial meningioma. Recurrent meningiomas are a clinical challenge and treatment at the time of recurrence is not well delineated.  Some concerns were noted as follows:

-the relatively small cohort sample (only 56 patients underwent a second operation for recurrence) in a long period (starting from 1988 when other current strategies, like SRS and gamma-knife radiosurgery were not available everywhere) is the major limitation and is made worse by it being a retrospective study.

-additional large-scale and prospective studies are required to better recognize the outcomes of reoperation surgery in meningioma.

However the study was well-written and rigorously conducted.

Author Response

Thank you for taking the time to review our manuscript investigating functional outcomes following repeat operation for meningioma. Please find below a summary of your concerns and how we have rectified them.

Concern: Small sample size and heterogeneity in treatment strategy compounded by retrospective nature

Response: We have acknowledged the sample size and treatment strategy issues in the limitations section, however we have now also noted the effect of the retrospective nature of the study.

Concern: Lack of proposal of future work in the area

Response: We have made a point to suggest that large-scale, multi-centre studies are needed to reach sufficient power to recognise outcomes in re-operated meningioma patients in the conclusion.

Thank you once again for your favourable review it is greatly appreciated and we hope these changes will sufficiently address your concerns.

Reviewer 2 Report

The authors sought to identify risk factors for a performance status outcome following a second operation for recurrent meningioma. They conducted a  retrospective, comparative cohort study and identified 712 patients had surgery for intracranial meningiomas between 1988 and 2018, of whom 56 (7.9%) underwent a second operation for recurrence. Primary outcome measure was World Health Organisation performance. Secondary outcomes were complications, and overall and progression free survival (OS and PFS respectively). Baseline clinical characteristics, tumour details, and operation details were collected. Multivariable binary logistic regression was used to identify risk factors for performance status outcome following second operation.

Fifteen patients (26.8%) had worsened performance status after second operation compared to three (5.4%) after the primary procedure (P=.002). An increased number of post-operative complications following second operation was associated with a poorer performance status following that procedure (odds ratio 2.2 [95% CI 1.1-4.6]). Second operation complication rates were higher than after the first surgery (46.4%, n=26 versus 32.1%, n=18, P=.069). Median OS was 312.0 months (95% CI 257.8-366.2). Median PFS following first operation was 35.0 months (95% CI 28.9-41.1). Following second operation, the median PFS was 68.0 months (95% CI 49.1-86.9).

The manuscript is well-written and well presented with informative graphs. It is well supported by data in the tables. It will be a valuable contribution to our science-based neurosurgical practice. The study period is extremely long (30 years), something the authors acknowledge as a limitation of their study, but which makes it is difficult to interpret their data and the relevance to modern meningioma surgery. 

I would like to see the authors elaborate (but keep it short) on the following:

1. How do the authors explain their finding that "The rate of Simpson grade 4 resections following first operation was higher in the 656 159 non-re-operated patients compared to the re-operated cohort (n=124, 18.9% versus n=31, 160 55.4%)"?

2. How do the authors explain their finding that "Median progression free survival following first surgery was 35.0 months (95% CI 236 28.9-41.1) and following second surgery was 68.0 months (95% CI 49.1-86.9)"? Patient selection or better surgery second time around? An effect of tumor location? (ref: Meningiomas: skull base versus non-skull base. Meling TR, et al. Neurosurgical Review 2019;42(1):163-173. DOI: 10.1007/s10143-018-0976-7) or the time factor? (ref: Meningiomas - are we making progress? Meling TR, et al. World Neurosurg 2019;125(5):e205-e213) doi.org/10.1016/j.wneu.2019.01.042

Section 3.2 misses information regarding the reoperated group and Simpson grade, as well as WHO - should be reported uniformly. 

The lack of MiB or Ki-67 should be mentioned under limitations (ref: Mirian C, et al. The Ki-67 Proliferation Index as a Marker of Time to Recurrence in Intracranial Meningioma. Neurosurgery 2020 (online ahead of print) DOI: 10.1093/neuros/nyaa226). 

Reference 29 is not related to meningioma surgery and should be omitted.

Lastly, some important references are missing: 

    1. Extent of Resection in Meningioma: Predictive Factors and Clinical Implications. Lemée JM, et al. Scientific Reports 2019;9(1):5944. DOI: 10.1038/s41598-019-42451-z
    2. Early postoperative complications in meningioma: predictive factors and impact on outcome. Lemée JM, et al. World Neurosurg 2019;128(8):e851-e858. doi.org/10.1016/j.wneu.2019.05.010
    3. Long-term 25-year follow-up of surgically treated parasagittal meningiomas. Pettersson-Segerlind J, et al. World Neurosurg. 2011;76(6):564-71. DOI: 10.1016/j.wneu.2011.05.015
    4. Bartek J Jr, et al. Predictors of severe complications in intracranial meningioma surgery: a population-based multicenter study. World Neurosurg. 2015;83(5):673-8. http://dx.doi.org/10.1016/j.wneu.2015.01.022 
    5. Corell A, et al. Neurosurgical treatment and outcome patterns of meningioma in Sweden: a nationwide registry-based study. Acta Neurochir (Wien). 2019;161(2):333-341. DOI: 10.1007/s00701-019-03799-3

Author Response

Many thanks for your detailed and thorough review of our manuscript investigating functional outcomes for patients following re-operation for recurrent intracranial meningioma. Please find below a summary of your concerns and how we have rectified them in the manuscript.

Concern: Explanation of difference in Simpson grades between re-operated and non-re-operated groups

Response: This section has been edited in the manuscript as the original submitted version contained an error. The re-operated group had a higher proportion of Simpson grade 4 resections compared to the non-re-operated group (as might probably be expected). The paragraph has now been changed to reflect this. Many thanks for highlighting this.

Concern: Explanation for the increase in progression free survival following second operation compared to first operation in the re-operated cohort.

Response: In the manuscript we have addressed this point by proposing that the difference is most likely due to a proportion of patients being “cured” from their re-operation, while every patient in this cohort will recur after first operation. We have expanded further upon this point in line with your suggestion, by suggesting that the increase in tumour control may occur due to surgeons aiming for greater extent of resection in the presence of a recurrent tumour.

Concern: Simpson grades and WHO grade are not included in the same section for the comparison between re-operated and non-re-operated cohorts.

Response: These datapoints have now been addressed in section 3.2

Concern: Lack of Ki-67 or MiB as a limitation of the study.

Response: This has now been addressed in the limitations of the study (with reference), as lack of Ki-67 could be an explanatory variable of the difference in progression free survival rates (Operation 1 versus Operation 2). However, since Ki67 is not part of WHO classification criteria it is only available for a minority of patients in our study and therefore there would be insufficient data to analyse.

Concern: Missing important references.

Response: All references have been added in manuscript.

We would once again like to thank you for taking the time to review the manuscript and suggest points for improvement. We hope that the changes we have made sufficiently address the concerns raised in your review.

Reviewer 3 Report

I read with interest the manuscript from Richardson et al., reporting the data of 712 patients operated from intracranial meningiomas from the eighties up to now.

Overall, the manuscript brings no new novelty, reporting rather old data. Recent reports with more recent data and twice the cohort have been published recently (see Lemee et al.)

The statistical methodology appears to be convoluted and suffers from several bias, the most obvious being the size disparity between the subgroups.  Furthermore, I really don’t get the statistical analysis, since either uni- or multivariate analysis should be performed, not both.

The graphs and figures are complicated and somewhat untidy.

Author Response

Thank you for taking the time to review our manuscript investigating functional outcomes of patients following re-operation for recurrent intracranial meningioma. We have addressed some of your concerns highlighted in your review with the summary below.

Concern: Manuscript brings no new novelty, with old data and small sample size

Response: Thank you for highlighting this issue. We feel that by including a functional performance score following re-operation we have added a new dimension to the topic. Our data comes from a wide time period with a high degree of treatment heterogeneity, both of which have been addressed within our limitations. With a low frequency event such as re-operation for recurrent meningioma, we feel that this is an unavoidable issue.

Concern: Issues surrounding statistical analysis.

Response: The statistical analysis plan was pre-defined and agreed in consensus with our research group. A binary logistical regression model was felt appropriate to analyse for factors associated with a worsening of performance status following second operation. To the knowledge of our authors, it is common practice to perform univariable analysis prior to multivariable, to ensure that redundant factors are not included in the multivariable model. We have included other univariable statistical tests however these serve a different purpose to the regression model. Tests of data distribution were included to ensure that data is appropriately presented. Kaplan Meier analysis was required to conduct survival analysis for the cohort, with log rank tests to conduct univariable analysis between progression free survival following operation 1 versus operation 2.

Concern: Cohort size bias.

Response: We acknowledge that there was a sizable difference in the cohort size between the re-operated and non-re-operated groups. This was felt to be acceptable since the extent of analysis was limited to descriptive statistics to describe the differences between the two cohorts. No further statistical analysis was conducted between the 656 patients in the non-re-operated group versus the 56 in the re-operated group. We also highlight that in the 2020 Lemée et al. study, there is a proportionally similar difference in non-re-operated versus re-operated patients (Lemée – 1290 vs 114 (8.1% re-operation rate), Richardson – 656 vs 56 (7.8% re-operation rate)).

Concern: Graphs and figures are untidy and complicated.

Response: Thank you for highlighting this issue. The cohort of patients in our study demonstrated a high degree of heterogeneity, which created issues when graphically illustrating the results. As a research group, we reached a consensus agreement on how best to illustrate our results (following multiple iterations of figures) and the graphs/diagrams presented in the manuscript were the result – we feel this is the best way to illustrate our data.

Many thanks for your thorough review of our proposed manuscript. We hope to have addressed your concerns in our response.